# The Inflammatory Cytokine IL-3 Hampers Cardioprotection Mediated by Endothelial Cell-Derived Extracellular Vesicles Possibly via Their Protein Cargo

**DOI:** 10.3390/cells10010013

**Published:** 2020-12-23

**Authors:** Claudia Penna, Saveria Femminò, Marta Tapparo, Tatiana Lopatina, Kari Espolin Fladmark, Francesco Ravera, Stefano Comità, Giuseppe Alloatti, Ilaria Giusti, Vincenza Dolo, Giovanni Camussi, Pasquale Pagliaro, Maria Felice Brizzi

**Affiliations:** 1Department of Clinical and Biological Sciences, University of Turin, Regione Gonzole 10, 10043 Orbassano, Italy; claudia.penna@unito.it (C.P.); stefano.comita@unito.it (S.C.); pasquale.pagliaro@unito.it (P.P.); 2Department of Medical Sciences, University of Turin, Corso Dogliotti 14, 10126 Turin, Italy; saveria.femmino@unito.it (S.F.); marta.tapparo@unito.it (M.T.); tatiana.lopatina@unito.it (T.L.); francesco.ravera@edu.unito.it (F.R.); 3Department of Biological Science, University of Bergen, Thormohlensgt 55, 5020 Bergen, Norway; kari.fladmark@uib.no; 4Uni-Astiss, Polo Universitario Rita Levi Montalcini, 14100 Asti, Italy; giuseppe.alloatti@unito.it; 5Department of Life, Health and Environmental Sciences, University of L’Aquila, Via Vetoio-Coppito 2, 67100 L’Aquila, Italy; ilaria.giusti@univaq.it (I.G.); vincenza.dolo@univaq.it (V.D.)

**Keywords:** cardiac damage, ischemia/reperfusion injury, endothelial-derived extracellular vesicles, inflammatory cytokines, interleukin 3

## Abstract

The biological relevance of extracellular vesicles (EV) released in an ischemia/reperfusion setting is still unclear. We hypothesized that the inflammatory microenvironment prevents cardioprotection mediated by endothelial cell (EC)-derived extracellular vesicles. The effects of naïve EC-derived EV (eEV) or eEV released in response to interleukin-3 (IL-3) (eEV-IL-3) were evaluated in cardiomyoblasts (H9c2) and rat hearts. In transwell assay, eEV protected the H9c2 exposed to hypoxia/reoxygenation (H/R) more efficiently than eEV-IL-3. Conversely, only eEV directly protected H9c2 cells to H/R-induced damage. Consistent with this latter observation, eEV, but not eEV-IL-3, exerted beneficial effects in the whole heart. Protein profiles of eEV and eEV-IL-3, established using label-free mass spectrometry, demonstrated that IL-3 drives changes in eEV-IL-3 protein cargo. Gene ontology analysis revealed that both eEV and eEV-IL-3 were equipped with full cardioprotective machinery, including the *Nitric Oxide Signaling in the Cardiovascular System*. eEV-IL-3 were also enriched in the endothelial-nitric oxide-synthase (eNOS)-antagonist caveolin-1 and proteins related to the inflammatory response. In vitro and ex vivo experiments demonstrated that a functional Mitogen-Activated Protein Kinase Kinase (MEK1/2)/eNOS/guanylyl-cyclase (GC) pathway is required for eEV-mediated cardioprotection. Consistently, eEV were found enriched in MEK1/2 and able to induce the expression of B-cell-lymphoma-2 (Bcl-2) and the phosphorylation of eNOS in vitro. We conclude that an inflammatory microenvironment containing IL-3 changes the eEV cargo and impairs eEV cardioprotective action.

## 1. Introduction

Cardiovascular diseases (CVDs), and in particular myocardial infarction (MI), are the most frequent causes of morbidity and mortality worldwide [1]. Primary percutaneous coronary intervention (PPCI) remains the most appropriate approach to restore the coronary blood flow and improve patients’ outcomes [2]. However, PPCI does not completely prevent patient overall mortality rate [3]. This implies that to limit myocardial ischemia/reperfusion (I/R) injury and improve clinical outcomes, new strategies should be investigated [4]. Many agents, acting at different sites of the damaging cascade, have been shown to reduce the infarct size in different I/R preclinical models [5]. However, these therapeutic options are still far from being approved in clinic [6]. The most relevant mechanisms accounting for the I/R injury include ATP depletion, the production of reactive oxygen species (ROS) and reactive nitrogen species (RNS), alterations of cell membrane permeability, loss of intracellular calcium homeostasis, and mitochondrial damage [7,8]. However, along with the direct effects of I/R, systemic inflammation via the release of pro-inflammatory cytokines can drive damaging cues to both vessels and cardiomyocytes [9,10]. It is worth noting that IL-3, originally described as a hematopoietic growth factor [11], has been recognized from among the inflammatory mediators [12] released by the activated T-lymphocytes into sites of inflammation [13]. Moreover, IL-3 induces endothelial cell (EC) activation and provides angiogenic cues both directly [14] and via paracrine mechanisms involving the release of extracellular vesicles (EV) [15]. These dual pro-inflammatory and proangiogenic IL-3 actions drove us to select IL-3 from among potential paracrine inflammatory mediators.

EV have recently been recognized as one of the most relevant mechanisms of cell-to-cell communication both at local and remote sites [15,16]. Indeed, EV contribute to several pathophysiological processes including angiogenesis, coagulation, tissue repair, and inflammation [16,17,18]. However, in models of I/R injury, the role of EV is still a matter of debate, since both beneficial and detrimental effects have been described [19]. Circulating EV collected after preconditioning protocols have been shown to significantly reduce I/R injury [20,21]. Yet, the activation of oxidative processes and the induction of cardiomyocyte apoptosis have also been reported upon treatment with EV released by EC (eEV) subjected to I/R. Moreover, increased local inflammation has been reported upon treatment with eEV released during acute myocardial infarction [22,23]. Additionally, intra-cardiac fibroblasts and macrophages contribute to the local inflammatory response during myocardial I/R injury [24,25,26]. These data suggest that the release of EV and the activation of the inflammatory response in the ischemic microenvironment may contribute to establish the degree of the overall damage. Therefore, the impact on the local response associated to I/R injury of naïve eEV and eEV released in an inflammatory setting containing IL-3 (eEV-IL-3) was evaluated using the in vitro and ex vivo myocardial I/R models. Moreover, to provide insight on the eEV and eEV-IL-3 mechanism of action, proteomic analysis was performed. Given the endothelial nature of EV, the role of the endothelial nitric oxide (NO) synthase (eNOS), the guanylyl cyclase (GC), and the Mitogen-Activated Protein Kinase-Extracellular Signal-Regulated Kinase (MEK1/2) pathways [27,28] have been investigated both in in vitro and in ex vivo models.

## 2. Materials and Methods

### 2.1. Culture of Human EC (Human Microvascular Endothelial Cell Line-1, HMEC-1 and Human Umbilical Vein Cells, HUVEC), and Rat Embryonic Cardiac Myoblasts (H9c2)

HMEC-1, HUVEC, and rat-derived H9c2 cells were obtained from the American Type Culture Collection (ATCC; Manassas, VA, USA). HMEC-1 were grown in MCDB 131 Medium supplemented with 10% fetal bovine serum (FBS), 10 ng/mL of epidermal growth factor, 1 μg/mL of hydrocortisone, 2 mM glutamine, and 1% (v/v) streptomycin/penicillin at 37 °C and 5% CO_2_, while Human Umbilical Vein Cells (HUVEC) were cultured in M199 plus 10% (v/v) bovine calf serum (BCS), 2% (v/v) of streptomycin/penicillin and 5 ng/mL of bFGF. H9c2 cells were grown at 37 °C and 5% CO_2_ in Dulbecco’s modified Eagle’s medium nutrient mixture F-12 HAM (DMEM-F12) and supplemented with 10% fetal bovine serum (FBS) and 1% (v/v) streptomycin/penicillin [29].

### 2.2. EC-EV Isolation and Characterization

To collect EV, overnight starved HUVEC were either left untreated or treated with IL-3 (10 ng/mL) or anti-IL-3Rα neutralizing antibody (R&D Systems, Minneapolis, MN, USA) as previously described [15]. The conditioned medium (40 pooled HUVEC cultures/EV group) was centrifuged for 30 min at 3000× *g* to remove cell debris and apoptotic bodies and then submitted to microfiltration with 0.22 μm filters (MF-Millipore) to remove larger vesicles. Then, the supernatant was ultracentrifuged for 2 h at 100,000× *g*, 4 °C, using the Beckman Coulter Optima L-100K Ultracentrifuge with the rotor type 45 Ti 45,000 rpm. The pellet of EV obtained was resuspended in DMEM supplemented with 1% DMSO. Then, the EV suspension (from 40 different sample collection) was collected as 3 individual pools and stored at −80 °C until further use. In selected experiments, 8 pooled HUVEC cultures were used to recover anti-IL-3R-eEV. EV were analyzed using NTA analysis, the NanoSight NS300 system (Malvern Instruments, Ltd., Malvern, PA, USA), and FACS analysis. EV flow cytometry analysis was performed using the MACSPlex Exosome Kit (human, Miltenyi Biotec Bologna, Italy), following the manufacturer’s protocol [30]. Each sample was analyzed using a CytoFLEX Flow Cytometer (Beckman Coulter, Brea, CA, US). CytExpert Software (Beckman Coulter, Brea, CA, USA) was used to analyze flow cytometric data. For EV quantification (particle number), NTA was used.

### 2.3. Transmission Electron Microscopy

After collection, EV were resuspended in PBS and properly diluted. Therefore, the samples were incubated for 5 min onto carbon-coated copper grids, 200 mesh (Electron Microscopy Sciences, Hatfield, PA, USA) at room temperature in a humidified chamber. The EV adsorbed on the grids were fixed (2% glutaraldehyde diluted in PBS) (Electron Microscopy Sciences, Hatfield, PA, USA) for 10 min, diluted in PBS for 10 min, washed 3 times in Milli-Q water, negative stained with 2% phosphotungstic acid, brought to pH 7.0 with NaOH, and air dried. Grids were observed using a Philips CM100 (Eindhoven, The Netherland) equipped with a detector at 80 kV, 64000X [31].

### 2.4. Hypoxia/Reoxygenation (H/R) Protocol

For the in-vitro hypoxia–reoxygenation (H/R) experiments, H9c2 cells were serum-starved (in exosome depleted FBS 2%) for 2 h. Cells were pretreated with different eEV, eEV-IL-3 number (7 × 10^3^, 1 × 10^4^ and 1 × 10^6^ EV/cell) or IL-3 (10 ng/mL) for 2 h and then exposed to hypoxia (1% O_2_, 5% CO_2_) for additional 2 h in the presence of eEV, eEV-IL-3, or IL-3 and subsequently reoxygenated (21% O_2_ and 5% CO_2_) for 1 h. For HMEC-1 and H9c2, co-culture 5 × 10^3^ HMEC-1 cells were seeded into each of Millicell 24-well plate inserts with 0.4 μm pores (Millipore, Milan, Italy) while 3 × 10^4^ H9c2 were seeded into the lower wells. Cells were allowed to attach for 24 h and serum-starved for additional 24 h. HMEC-1 were treated or not with eEV, eEV-IL-3, or IL-3 (10 × 10^4^/cell for eEV and eEV-IL-3 (found effective in preliminary dose response curve) and 50/ng/mL for IL-3) for 2 h, exposed to hypoxia (1% O_2_, 5% CO_2_) for an additional 2 h, and subsequently re-oxygenated (21% O_2_ and 5% CO_2_) for 1 h. H9c2 viability was evaluated with 3-(4,5-dimethylthiazol-2-yl)-2,5-diphenyltetrazolium bromide (MTT) assay [29]. In selected experiments the above-indicated stimuli were also used in combination with U0126 (15 µM), L-NNA (100 µM) or ODQ (100 µM) [32,33,34].

### 2.5. MTT Assay

At the end of the H/R protocol, cell viability was assessed using the 3-(4,5-dimethylthiazol-2-yl)-2,5-diphenyltetrazolium bromide (MTT) kit (10 µL/well) as indicated by manufacturers. Briefly, after 2 h incubation at 37 °C, dimethyl sulfoxide (DMSO) was added. The plates were read at 570 nm to obtain the optical density values [29].

### 2.6. Label-Free Mass Spectrometry Analysis

Protein (20 µg) denaturation, reduction, and trypsination and peptide up-concentration and desalting were performed as in [35]. Tryptic peptides were dissolved in 5% ACN, 0.1% TFA were injected into an Ultimate 3000 RSLC system (Thermo Scientific, Sunnyvale, CA, USA) connected online to a Q-Excative HF mass spectrometer (Thermo Scientific, Bremen, Germany) and equipped with EASY-spray nano-electrospray ion source (Thermo Scientific, Monza, Italy).

Peptide trapping and desalting in the nano-LC were as described in [35]. Peptide separation (120 min) was also as in [35] apart from the following change in settings: (1) solvent A and B were 0.1% TFA (v/v) in water and 100% ACN respectively, (2) gradient composition was 5% B during trapping (5 min) followed by 5–8% B over 0.5 min, 8–35% B for the next 134.5 min, and 35–90% B over 15 min. MS spectra were acquired as in [35] apart from a change in intensity threshold to 5e4 and resolution to R = 60,000. The raw files were searched in MaxQuant (version 1.5.2.8, Heidelberg, Germany) against a human reviewed database from UniProt (downloaded November 2015, Cambridge, UK), The following search parameters were used; carbamidomethyl (Cys) as fixed modification, oxidation (Met), phosphorylation (STY), and acetyl (Protein N-terminal) as variable modifications. We allowed for two miss cleavages of trypsin, 20 ppm for precursors, and 0.6 Da for fragment ion mass tolerance, and a false discovery rate (FDR) at 1%. Only unmodified and two unique + razor peptides were used for quantifications. Perseus (1.5.2.6, Heidelberg, Germany) was used for further analysis. Proteins identified by site, reverse hit, and potential contaminants were removed. The protein list was further reduced by only keeping proteins with at least 3 out of 6 valid values. The LFQ intensity values were log2 transformed, and proteins were considered significant if they passed the two sample t-test with the following settings; S0 = 2 and FDR 0.01. *p*-values were given as Log10 value.

### 2.7. Protein Pathway Analysis

A list of unique proteins identified by MS/MS spectra and significantly expressed was taken into consideration to performed GO analysis and pathway enrichment analysis. Gene ontology analysis was performed with Funrich V3 software looking at molecular function and cellular process category [36]. Pathway enrichment analysis and disease association analysis were performed using Ingenuity Pathway Analysis (IPA) software (Qiagen, Milan, Italy).

### 2.8. Animals

Male Wistar rats (4–5 months old, body weight 400–450 g, Harlan Laboratories, Udine, Italy) were used for the assessment of cardio-protection as below specified in the I/R studies. Rats received human care according to the European Directive 2010/63/EU on the protection of animals used for scientific purposes. The Italian National Institute of Health Guide for the Care and Use of Laboratory Animals (protocol no: E669C.N.OVL) approved the animal protocols. Animals were housed under controlled conditions with free access to tap water and to standard rat diet.

### 2.9. Ischemia/Reperfusion (I/R) Studies

Rats were anesthetized and heparinized (800 U/100 g b.w., i.m.). Depth anesthesia was assessed by the absence of withdraw reflexes. Then, hearts were quickly excised, placed in ice-cold buffer solution, cleaned up, and weighed. Hearts were attached to the perfusion apparatus and perfused with oxygenated Krebs–Henseleit buffer solution (KHS) through a cannula inserted into the ascending aorta. The KHS contained (mM): 127 NaCl, 5.1 KCl, 17.7 NaHCO_3_, 1.26 MgCl_4_, 1.5 CaCl_2_, 11 D-glucose (pH 7.4; 37 °C; 95% O_2_/5% CO_2_). The hearts were located in a temperature-controlled chamber (37 °C). Cardiac preparations were perfused in constant-flow mode. In order to assess the cardiac preparation, coronary perfusion and left ventricular pressure were monitored during the entire experiments [37].

### 2.10. Experimental Groups

To verify the protective effect of eEV and eEV-IL-3, the hearts were assigned to one of the experimental groups below described. In all groups, except for SHAM, the hearts were subject to I/R protocol consisting of 30 min stabilization and 30 min of norm-thermic global ischemia; then, the ischemia was followed by 60 min reperfusion. eEV and eEV-IL3 doses were selected based on our preliminary experiments (data not shown).

(1)SHAM (*n* = 3) only KHS has been infused.(2)I/R group (*n* = 8) after stabilization, only I/R protocol was performed [38].(3)eEV group (*n* = 5), eEV (1 × 10^9^/mL final concentration) were diluted in KHS and infused into the hearts, through a collateral line for 10 min, then the hearts underwent I/R protocol.(4)eEV-IL-3 group (*n* = 5) eEV-IL-3 (1 × 10^9^/mL final concentration) were diluted in KHS and infused into the heart, through a collateral line for 10 min, then hearts underwent I/R protocol.(5)IL-3 group (*n* = 5) IL-3 (50 ng/mL) [14] was diluted in KHS and infused into the heart, through a collateral line for 10 min, then the hearts underwent I/R protocol.(6)Triton X-100 (*n* = 3), in these hearts, the endothelium was made dysfunctional by a 0.1 mL Triton X-100 (5 μL/mL) injection [39,40].(7)eEV after Triton X-100 (*n* = 3), in these hearts, eEV (1 × 10^9^/mL final concentration) infusion was performed after the endothelium was made dysfunctional by a 0.1 mL Triton X-100 (5 μL/mL) injection [39,40].(8)eEV+L-NNA group (*n* = 3), the eNOS inhibitor N omega-nitro-L-arginine (LNNA, 100 µM) was used to assess the involvement of eNOS enzyme in eEV-induced cardioprotection (1 × 10^9^/mL final concentration) [41].(9)eEV+U0126 group (*n* = 3), the MEK1/2 blocker 1,4-Diamino-2,3-dicyano-1,4-bis(2 aminophenylthio) butadiene (U0126: 60 µM) was used to ascertain the involvement of MEK1/2 in eEV-(1 × 10^9^/mL final concentration) induced cardioprotection [38].(10)eEV+ODQ group (*n* = 3), the GC blocker 1H-(1,2,4)oxadiazolo(4,3-a)quinoxalin-1-one (ODQ;10µM) was used to ascertain the involvement of the GC enzyme in eEV- (1 × 10^9^/mL final concentration) induced cardioprotection [37].

### 2.11. Infarct Size Assessment

The extent of infarct size was evaluated using a gravimetric method in a blind manner. At the end of each experiment, the hearts were quickly removed from the perfusion apparatus and the left ventricle was dissected into 2–3 mm circumferential slices. After 20 min of incubation at 37 °C in a 0.1% solution of nitro-blue tetrazolium in phosphate buffer, the colored vital tissue was carefully separated from the non-stained necrotic tissue and weighed. Since ischemia is global, the entire total left ventricular mass was considered as the area at risk, and the necrotic mass was expressed as percentage of the total ventricle mass [38].

### 2.12. Western Blot Analysis

eEV and eEV-IL-3 were lysed in lysis buffer (RIPA buffer with proteinase inhibitors). Starting from the same EV particle number, protein samples were quantified by the Bradford method before performing Western blot (50 µg proteins/each sample were loaded). Anti-CD63, anti-CD81, antiHSP90, anti-GM130, anti-Bcl-2 (Abcam, Milan, Italy), anti-MEK1/2, (Cell Signaling, Danvers, MA, USA), anti-CD29, anti-caveolin-1, anti-p-eNOS-ser1177 (Invitrogen, Carlsbad, CA, USA), and anti-vinculin (Millipore, Milan, Italy) antibodies were used as primary antibodies. Appropriate HRP-conjugated secondary antibodies (BioRad, Milan, Italy) were used, and proteins were detected with Clarity Western ECL substrate (BioRad, Milan, Italy). Image Lab Software (BioRad Milan, Italy) instrument was used for densitometric analysis. Data are expressed as arbitrary unit. Ponceaus-staining has been used as input (EV protein content normalization).

### 2.13. Chemicals

The sources of the specific antibodies are shown in the different sections. If not differently specified, the other reagents were obtained from Sigma (St. Louis, MO, USA).

### 2.14. Statistical Analysis

Data are expressed as mean ± SEM. Statistical analysis was performed as follows: *t* test was used for the comparison between two groups, while one-way ANOVA was used for comparison between ≥3 groups followed by Student’s *t*-test or the Newman–Keuls multiple-range test depending on the experiments. *p*-value ≤ 0.05 was considered as significant. Graph Pad Prism version 8.2.1 (Graph Pad Software, Inc, USA) was used for statistical analyses.

## 3. Results

### 3.1. eEV and eEV-IL-3 Have Similar Size and Surface Markers

eEV and eEV-IL-3 were first recovered from naïve Human Umbilical Vein EC (HUVEC) and HUVEC exposed to IL-3 and subjected to NanoSight, transmission electron microscopy (TEM) and FACS analyses. Since cells may undergo apoptosis or autophagy in the absence of serum and release apoptotic bodies, EV obtained after filtration were used. As previously reported [15], IL-3 treatment boosts the release of EV by HUVEC; however, no difference between eEV and eEV-IL-3 size was detected by Nanosight (193 ± 4.3 nm and 192.4 ± 3.5 nm respectively) and TEM (Figure 1A). As expected, endothelial markers (CD31, CD146, and CD105) were found highly expressed in both eEV and eEV-IL-3 (Figure 1B). Western blot analysis was performed to validate the expression of exosomal markers (Figure 1B). GM130 protein served as a negative marker of EV (Figure 1B). Moreover, FACS analysis, using the MACSPlex exosome kit, revealed a similar pattern of surface marker expression (Figure 1C). Endothelial and exosomal markers identified in proteome analysis are reported in Figure 1D.

### 3.2. Both eEV and eEV-IL-3 Induce Protection in a Simulated In Situ Condition, While Only eEV Directly Trigger Cardioprotective Signals

First, the biological effects exerted by eEV and eEV-IL-3 were evaluated on primary cultures of cardiomyoblasts (H9c2) exposed to the in vitro model of hypoxia/reoxygenation (H/R) (Figure 2A). IL-3 served as an internal control. As shown in Figure 2B,C, H9c2 cell death was prevented in response to eEV, but not eEV-IL-3 or IL-3. In order to mimic the in vivo condition, experiments were also performed in transwells. To this end, HMEC-1 were seeded in the upper chamber and pre-conditioned for 2 h with eEV and eEV-IL-3. H9c2 were seeded in the lower chamber. Then, the cells were subjected to the H/R protocol (Figure 2A,D) and H9c2 viability was analyzed. As shown in Figure 2E, eEV significantly improved cell viability, not only with respect to untreated hypoxic cells (NONE), but also when compared to normoxic conditions (CTRL N). A slight increase of H9c2e cell viability was also detected upon eEV-IL-3 treatment. IL-3 alone had no protective effects (Figure 2E).

### 3.3. eEV, But Not eEV-IL-3, Exert Endothelial-Dependent Protection against I/R in the Whole Heart

Since the infarct size (as percentage of AAR) is considered a clearer indicator of I/R injury, AAR was evaluated in our experimental groups [42]. Figure 3A shows the ex vivo I/R protocol and treatment (1 × 10^9^ eEV or eEV-IL-3) we selected based on our preliminary studies (data not shown). The hearts infused with Krebs–Henseleit solution (KHS) (SHAM) served as internal control. In the I/R control group, the hearts developed infarction, which corresponded to 60 ± 2% of area at risk (AAR). Of note, the infarct size was 43 ± 3% of AAR in the hearts pretreated with eEV (Figure 3B,C; *p* < 0.001 eEV vs. I/R group). This protective effect was not observed in the hearts that have been exposed to eEV-IL-3 or IL-3 (Figure 3B,C). Indeed, an infarct size (61 ± 3% of AAR for eEV-IL-3, and 72 ± 5% of AAR for IL-3; Figure 3B,C) similar to the control group was noticed in these groups. To evaluate whether the integrity of the EC monolayer could impact on tissue outcomes, ex vivo experiments were extended to the hearts, which have been pretreated with Triton X100 to induce EC damage. As shown in Figure 3B,C, treatment with eEV was no more effective upon EC damage.

### 3.4. eEV Are Enriched in MEK1/2 and HPS90 While eEV-IL-3 in the eNOS Antagonist, Caveolin 1

To gain further insight into the mechanisms accounting for the ex vivo results, protein profiling of eEV and eEV-IL-3 was performed. Label-free mass spectrometry analysis identified a total of 2077 proteins (Figure 4A). Of these, 736 proteins met the criteria for further statistical analysis based on the identification of two or more unique peptides. Of these, 651 significant proteins were carried by both eEV and eEV-IL-3, while 54 were carried by eEV-IL-3 and 8 by eEV. Detailed information of the differentially expressed proteins, Student *t*-test statistical analysis, and their respective ratio are reported in Appendix A. Ingenuity Pathway Analysis (IPA analysis) demonstrated that most of the proteins carried by eEV and eEV-IL-3 are related to cardiovascular diseases and EC remodeling, and have eNOS as a downstream effector (Table 1 and Table 2). These data were validated by analyzing the protein cargo of EV released by EC exposed to IL-3 and pretreated with an anti-IL-3R blocking antibody (anti-IL-3R-eEV) (Figure 4B). As shown in Figure 4B, almost all the up- or down-regulated proteins in eEV-IL-3 (vs. eEV) moved back to the eEV content in anti-IL-3R-eEV, indicating a specific protein content associated with IL-3 challenge. Pathway analysis of the eEV-IL-3 content showed the enrichment of *oxidative stress-related pathways, ER stress, calcium signaling,* and *VEGF pathway* (Table 1). Moreover, among the enriched pathways, *Nitric Oxide Signaling in the Cardiovascular System* was also found (Figure 4C, Table 1). In particular, a full cardioprotective machinery, including mitochondrial enzymes, control of cell-to-cell contact, cytoskeleton, and proteins involved in NO production was found in eEV-IL-3 (Table 1). Of interest, among proteins differentially expressed, we found an enrichment in the eNOS antagonist, caveolin 1 (Appendix A, Figure 4D) in eEV-IL-3. Moreover, we found that eEV-IL-3 was enriched in proteins involved in the activation of the inflammatory response (Table 3). In particular, proteins involved in the activation of immune cells, as well as in the IL-8 and nuclear factor kappa B (NFκB) pathways were found enriched in eEV-IL-3 (Table 3, Appendix A). Moreover, proteins classified in the IL-8 and NFκB pathways, and commonly involved in the signaling of different interleukins (IL-2, IL-6, IL-15, IL-17, and others) were also increased in eEV-IL-3 (Table 3, Appendix A). The only proteins enriched in eEV were MEK1/2 and HSP90 (Appendix A, Figure 4D). This suggests that they can contribute to eEV-mediated cardioprotection.

To evaluate whether eEV cargo may specifically reflect the micro environmental cues driving their release, eEV-IL-3 protein cargo was also compared to EV released by HUVEC in response to TNF-α (eEV-TNF-α) [43]. As shown in Appendix A, only a few proteins are shared by eEV-IL-3 and eEV-TNF-α.

### 3.5. MEK1/2/eNOS/GC Pathway Is Involved in eEV-Mediated Cardio-Protection

Since MEK1/2 was enriched while caveolin 1 reduced in eEV, and only eEV were found effective in inducing cardioprotection ex vivo, the MEK1/2/eNOS/GC pathways were investigated both in vitro and in the whole heart. To mimic the ex vivo experiments, transwells experiments were first performed using different MEK1/2/eNOS/GC inhibitors. As reported in Figure 5a, pretreatment with the MEK1/2 (U0126), eNOS (L-NNA), and GC (ODQ) inhibitors prevented eEV-mediated protection. Ex vivo experiments were also performed to corroborate these results. As shown in Figure 5b all inhibitors were effective in impairing eEV-mediated ex-vivo cardioprotection (61 ± 1%, 65 ± 1%, and 63 ± 3% respectively). None of the inhibitors had effects when used alone neither in vitro nor ex vivo (Appendix A).

### 3.6. eEV But Not eEV-IL-3 Treatment Induces the Expression Bcl-2 and the Phosphorylation of eNOS In Vitro

To validate the above results, the expression of MEK1/2 and the phosphorylation of eNOS were evaluated in vitro. The results reported in Figure 6A clearly demonstrate that eEV, unlike eEV-IL-3, are able to induce the expression of MEK1/2 and the phosphorylation of eNOS in H9c2 cells in co-cultures. The phosphorylation of eNOS was also detected upon a direct stimulation of H9c2 cells with eEV. On the contrary, MEK1/2 expression was not significantly increased upon eEV treatment (Figure 6B). The high basal MEK1/2 expression in H9c2 cells directly subjected to H/R may possibly explain these results.

Since bioinformatics analysis indicated that eEV-IL-3 are enriched in proteins involved in the IL-3 signaling pathways, and among them, a number of these proteins linked to the apoptosis signaling were found (Table 1), the expression of the anti-apoptotic protein Bcl-2 was also evaluated both in co-culture and upon direct stimulation. As shown in Figure 6A,B, eEV, unlike eEV-IL-3, significantly increased the expression of Bcl-2. These results confirm the ability of eEV to trigger anti-apoptotic cues and straighten the pro-apoptotic action of eEV-IL-3 in H/R setting.

## 4. Discussion

In the present study and consistent with our hypothesis, we demonstrated that the inflammatory microenvironment, recapitulated by eEV-IL-3, prevents cardioprotection. Of note, cardioprotective effects against I/R damage were detected upon eEV treatment. Moreover, we noticed that both the integrity of the endothelial layer and functional MEK1/2/eNOS/GC pathways are crucial for eEV-mediated cardioprotection.

EV can be released by different cell types and contribute to cell-to-cell communication either at local or distant site both in physiological and pathological settings [15,16,44]. In the heart, the ischemic damage induces the release of EV from cells of different origin [19,45]. Actually, it has been extensively reported that cells in the ischemic microenvironment release EV, which differ in term of composition and impact on target cells [45]. The biological relevance of eEV in the activation of signals leading to cardiovascular damage and/or tissue regeneration has been extensively documented [19]. Hypoxia associated to the ischemic injury primarily drives changes in EC and in their released eEV, particularly in their non-genetic and genetic cargo [22,23]. However, shift in the eEV molecular content in vivo can also result from the inflammatory response elicited by resident and recruited inflammatory cells [46]. In turn, this may drive the activation of damaging cues as well as long-lasting regenerative programs in the ischemic tissue [46]. Since the effects of EV released from EC in response to the inflammatory stimuli have been poorly investigated, we aimed to evaluate whether and how EV released by EC exposed to the inflammatory cytokine IL-3 impact on cardioprotection. Originally, IL-3 was described as a hematopoietic cytokine acting on progenitors and mature cells [11]. However, it has been also shown that IL-3 can functionally activate EC and can induce angiogenesis [14]. Apart from the direct effects on EC, recent evidence demonstrated that IL-3 can also act as paracrine mediator via EV [15]. Davidson et al. [47] have shown that eEV released from normoxic EC protect cardiomyocytes from H/R. Consistently, we noticed that eEV, unlike eEV-IL-3, protect H9c2 cells against H/R injury. Moreover, to mirror the physiological setting in which a close interaction and cross-talk between EC and cardiomyocytes dictate the response to injury, transwell experiments were also performed. We demonstrated that, although less effective than eEV, eEV-IL-3 preconditioning reduced the percentage of cardiac cell death, suggesting that, even in response to the inflammatory stimulus, EC, in vitro stimulated, release EV enriched in a cardioprotective machinery. As a matter of fact, proteomic analysis comparing eEV and eEV-IL-3 demonstrated that both eEV and eEV-IL-3 are equipped with proteins retaining cardioprotective properties. More importantly, we showed that most of these proteins were enriched in eEV-IL-3. Pretreatment of EC with an anti-IL-3R antibody before IL-3 stimulation provide evidence that the enriched proteins strictly depended on IL-3 challenge. Surprisingly, but in line with the results obtained when H9c2 cells were directly stimulated with EV, in the ex vivo experiments, only eEV protected the myocardium from I/R injury.

During I/R injury, the loss of EC physiological functions impacts on tissue outcome [7,8]. Our finding that pretreatment of EC with Triton prevented eEV-mediated protection supports the notion that a preserved endothelium is required for eEV effectiveness. miRNA EV cargo has been reported to induce cardioprotection in several models [18]. However, while eEV-IL-3 were found enriched in proangiogenic miRNAs [15], no protective effects were detected ex vivo. The short ex vivo exposure to eEV-IL-3 during the I/R protocol may explain their failure to activate the angiogenic switch via miRNA transfer. Nevertheless, to gain insight on eEV and eEV-IL-3 discrete biological actions, the entire EV cargo was deeply analyzed. In particular, we found that eEV-IL-3, besides being enriched in proteins associated with the oxidative stress related pathways, ER stress, Calcium signaling, VEGF, and the Nitric Oxide Signaling pathway, is also enriched in the main coat protein of caveolae, caveolin 1. It has been extensively reported that caveolin 1 impacts on oxidative stress and cardiac disease by controlling eNOS activity [48]. eNOS is mainly expressed by EC and heart myocytes [49]. After association with caveolae, eNOS interacts with caveolin 1 in EC and caveolin 3 in cardiac myocytes [50] and becomes inactive. Consistently, it has been reported that in the diabetes setting, the eNOS-derived NO release can be improved by inhibiting the binding of eNOS to caveolin 1 [51]. Similarly, the reduced cardioprotective effect of ischemic preconditioning in diabetic rats has been linked to the up-regulation of caveolin 1 [52]. These observations sustain the possibility that the lack of effectiveness of eEV-IL-3, rather than in the transfer of the entire cardioprotective machinery, may rely on a more efficient caveolin 1 binding to eNOS, holding eNOS in an inactive conformation. This hypothesis has been validated by our in vitro experiments showing that eEV, unlike eEV-IL-3, triggers eNOS phosphorylation. Alternatively, since eEV-IL-3 were also found enriched in inflammation-related proteins, their transfer into resident cardiac macrophages or fibroblasts in the ischemic myocardium may induce a rapid stimulation of the inflammatory response [25,26,53,54], which translates into the local activation of death signals preventing protection. A detrimental role of NFκB in preconditioning-induced protection against I/R has been also reported [55,56]. We found that along with proteins associated with the Acute Phase Response Signaling, the IL-1 and IL-8 pathways, a number of proteins related to nuclear NFκB were found enriched in eEV-IL-3. We also noticed that eEV-IL-3 were equipped with many proteins generally associated with the IL-3 pro-inflammatory signaling pathway, thus suggesting that the pro-inflammatory properties of IL-3 can be transferred to eEV-IL-3. Finally, data obtained by comparing eEV-IL-3 and eEV-TNF-α protein content [43] further sustain the notion that the local microenvironment dictates the unique eEV cargo.

Several endogenous protective mechanisms are known to be activated in the heart upon I/R, even via EV [55]. Essentially, different molecules including eNOS, PI3-kinase/Akt, ERK1/2, protein kinase C, STAT, and many others were found involved in preconditioning protective signaling [27,28]. In the present study, we found MEK1/2, HSP90, and caveolin 1 among proteins differentially expressed in eEV and eEV-IL-3, and that the activation of MEK1/2/eNOS/GC pathways by eEV is relevant for their cardioprotective action. Moreover, since the HSP90 acts to protect cells against stress-induced injury [57], its enrichment in eEV may trigger anti-apoptotic signals. Consistent with this possibility, Bcl-2 expression was increased in cells treated with eEV. Therefore, our data suggest that the selective activation of specific signaling components may elicit eEV-mediated cardioprotection (Figure 7). However, since the EV mechanism of action strictly relies on their entire molecular cargo, additional molecules carried by eEV and eEV-IL-3 may dictate their discrete biological actions.

Overall, this study demonstrated that IL-3 released in an inflammatory setting drives changes in eEV-IL-3 protein cargo that may contribute to the lack of cardioprotection (Figure 7). Different hypotheses can be postulated to explain the failure of eEV-IL-3 to induce protection: (a) the transfer of caveolin 1, which harms eNOS activity; (b) the transfer of inflammatory mediators activating death signaling in cardiomyoblasts, and (c) the activation of resident macrophages/fibroblasts boosting the inflammatory cascade [25]. The relevance of the inflammatory response associated with eEV-IL-3 challenge is supported by the bioinformatics analysis and by the observation that IL-3, itself, induced a powerful damaging effect. Nevertheless, our data shed light on eEV and eEV-IL-3 protein cargo and provide evidence for the ability of eEV to convey cardioprotective factors in the whole heart. Finally, our data suggest that the enrichment of MEK1/2 and HSP90 as well as the reduced expression of caveolin 1 in eEV may account for their mechanism of action.

## Figures and Tables

**Figure 1 cells-10-00013-f001:**
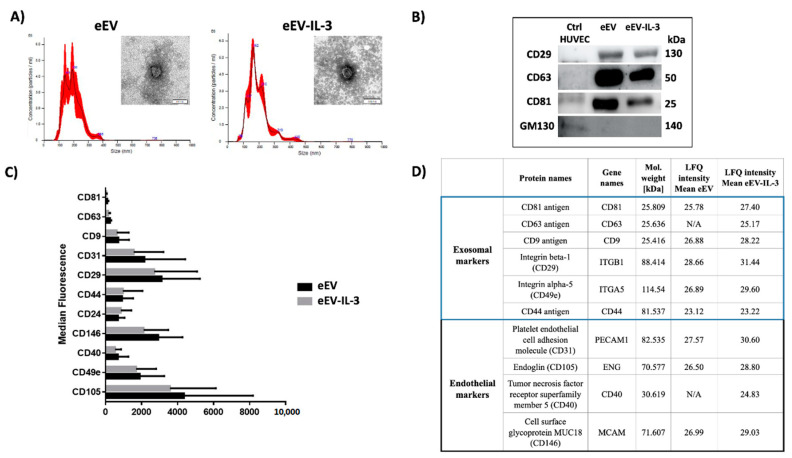
Endothelial cell-derived extracellular vesicles (eEV) and eEV released in response to interleukin-3 (eEV-IL-3) characterization. (**A**) Representative images of NanoSight analyses and TEM performed on eEV and eEV-IL-3. (**B**) Western blot. Representative images of exosomal markers, CD29, CD63, and CD81, expressed in eEV and eEV-IL-3. GM130 protein expression served as a negative EV marker. Indeed, it was expressed only in the cell extract (Ctrl Human Umbilical Vein Cells (HUVEC)). (**C**) eEV and eEV-IL-3 characterization with MACSPlex Exosome Kit. (**D**) The table reports the expression of eEV and eEV-IL-3 exosomal and endothelial markers provided by mass spectrometry.

**Figure 2 cells-10-00013-f002:**
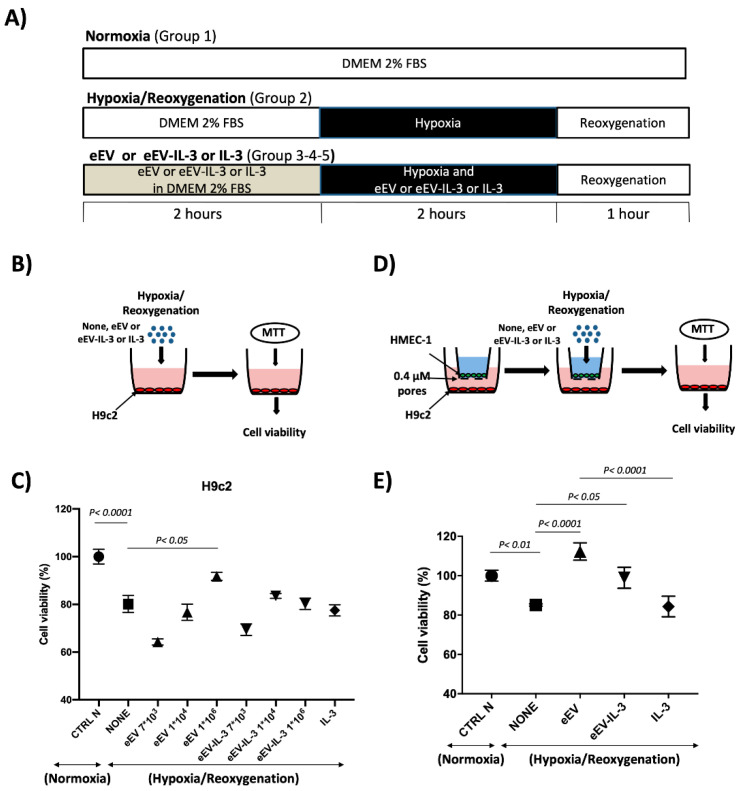
eEV, unlike eEV-IL-3, directly protect H9c2 cells from hypoxia/reoxygenation (H/R) injury. (**A**) Timeline of the in vitro protocol. Protocols of hypoxia (black boxes) and reoxygenation (white boxes) of H9c2 cells and EC. eEV-IL-3, eEV, or IL-3 were given for 2 h before hypoxia and during hypoxia. Cells were subjected to reoxygenation for 1 h. (**B**) Representative image of H9c2 cell treatment. (**C**) Cell viability of H9c2 cells exposed to normoxia (N) and hypoxia/reoxygenation (H/R) conditions in the presence of the indicated number of eEV and eEV-IL-3/cell. Data are presented as percentage variation with respect to mean value of cell count in normoxia (CTRL N) (*n* = 3). *p* < 0.0001 CTRL N vs. NONE; *p* < 0.05 NONE vs. eEV 1 × 10^6^. (**D**) Representative image of trans-well assay of Human Microvascular Endothelial Cell Line-1 (HMEC-1) cells seeded in the upper well and H9c2 cells in the lower well. HMEC-1 cells treated or not with eEV or eEV-IL-3 were subjected to H/R conditions. 3-(4,5-Dimethylthiazol-2-yl)-2,5-diphenyltetrazolium bromide (MTT) assay was performed on H9c2 cells. (**E**) Cell viability of H9c2 cells exposed to H/R conditions (*n* = 4). *p* < 0.01 CTRL N vs. NONE and eEV; *p* < 0.0001 NONE vs. eEV and eEV vs. IL-3. Data were normalized to the mean value of normoxic control (CTRL N).

**Figure 3 cells-10-00013-f003:**
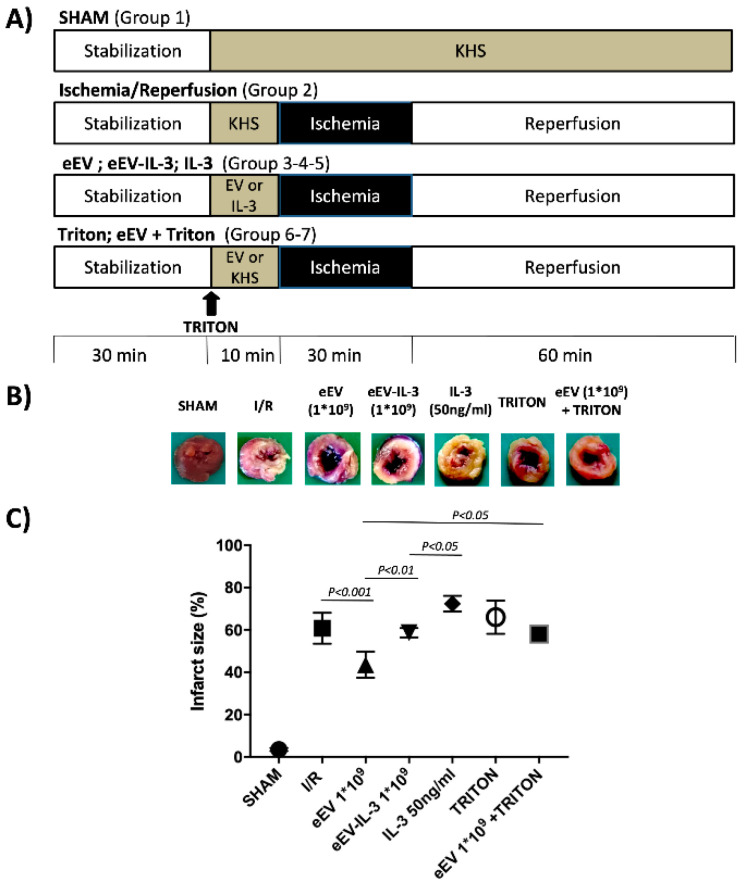
eEV, but not eEV-IL-3, protects isolated rat hearts from I/R-induced damage. (**A**) Timeline *of ex vivo protocol*. Protocols of ischemia (black boxes) and reperfusion (white boxes) in isolated rat hearts. After stabilization (white boxes), Krebs–Henseleit buffer solution (KHS) was infused alone (I/R) or with eEV, eEV-IL-3, or IL-3. eEV (1 × 10^9^), eEV-IL-3 (1 × 10^9^), or IL-3 (50 ng/mL) given for 10 min before ischemia, while Triton before KHS or eEV. The KHS-infused hearts not subjected to I/R (SHAM) served as internal control. (**B**) Representative images of the heart slices, obtained after NBT staining. The images correspond to SHAM, I/R, eEV 1 × 10^9^, eEV-IL-3 1 × 10^9^, IL-3 (50 ng/mL), and TRITON eEV 1 × 10^9^ + TRITON, respectively. (**C**) Infarct size in isolated rat hearts exposed to 30 min of ischemia plus 60 min of reperfusion, pretreated or not with eEV, eEV-IL-3, or IL-3 10 min before inducing ischemia. The amount of necrotic tissue measured after I/R protocol is reported as percentage of the left ventricle mass (LV; % IS/LV) for I/R (*n* = 8), eEV 1 × 10^9^ (*n* = 5), eEV-IL-3 1 × 10^9^ (*n* = 5), IL-3, 50 ng/mL (*n* = 5), TRITON (*n* = 3) and eEV 1 × 10^9^ +TRITON (*n* = 3). *p* < 0.05 eEV vs. TRITON and IL-3 vs. eEV-IL-3; *p* < 0.01 eEV vs. eEV-IL-3; *p* < 0.001 I/R vs. eEV.

**Figure 4 cells-10-00013-f004:**
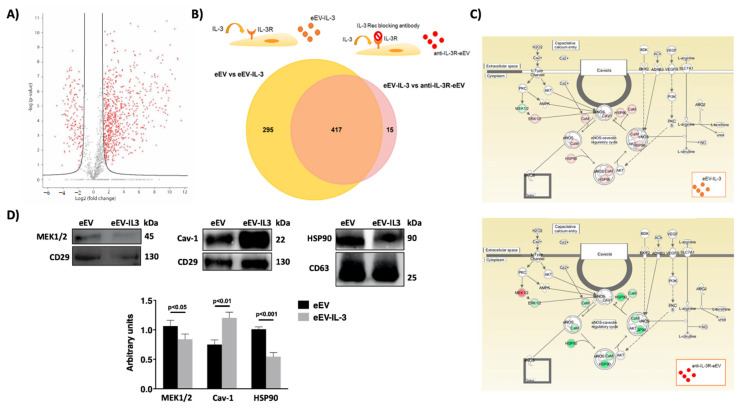
Protein profiling of eEV-IL-3 compared to eEV. (**A**) Volcano plot of IL-3 induced protein changes by label-free quantitation. Protein expression ratio of eEV-IL3/eEV (log2 scale) in label-free quantitation plotted against the -log of two-sample *t*-test. The lines represent threshold values (*p*-value 0.05, Fold Change >2). (**B**) Venn diagram. Venn diagram shows proteins in eEV-IL-3 compared to eEV (yellow) and anti-IL-3R-eEV (dark pink). Proteins included in the light pink area correspond to proteins only carried by anti-IL-3R-eEV. (**C**) nitric oxide signaling in the cardiovascular system. In the scheme, proteins modulated in eEV-IL-3 vs. eEV (upper panel) and homecoming in anti-IL-3R-eEV (lower panel) are highlighted. Red corresponds to increased expression; green corresponds to reduced expression. (**D**) Western blot analysis of eEV and eEV-IL-3. Representative image and relative quantification of MEK1/2 and the negative eNOS regulator, caveolin-1, HSP90 normalized to CD63 and to CD29 respectively (an aliquot of the three pooled samples were used) (MEK1/2: *p* < 0.05 eEV vs. eEV-IL-3; Caveolin-1: *p* < 0.01 eEV vs. eEV-IL-3; HSP90: *p* < 0.001 eEV vs. eEV-IL-3).

**Figure 5 cells-10-00013-f005:**
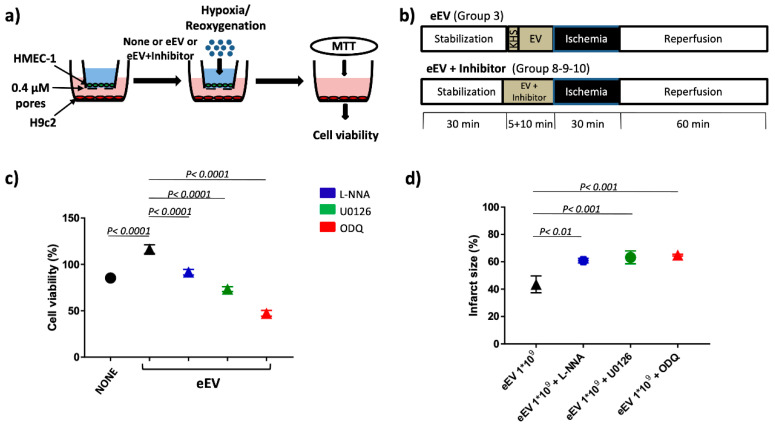
Mitogen-Activated Protein Kinase Kinase (MEK1/2)/endothelial-nitric oxide-synthase (eNOS)/ guanylyl-cyclase (GC) inhibitors prevented cardio-protection in in-vitro and ex-vivo models. (**a**) Representative image of transwell assay of HMEC-1 cells seeded in the upper and H9c2 cells in the lower well. HMEC-1 cells treated or not with eEV or eEV-IL-3 were subjected to H/R condition. MTT assay was performed on H9c2 cells. (**b**) Timeline of ex vivo protocol. Protocols of ischemia (black boxes) and reperfusion (white boxes) in isolated rat hearts. After stabilization (white boxes), Krebs–Henseleit buffer solution (KHS) was infused with eEV and/or inhibitors (gray boxes) via a collateral line. eEV (1 × 10^9^) for 10 min before ischemia, while inhibitors were given 5 min (total time infusion was 15 min) as indicated. (**c**) Cell viability of H9c2 cells subjected to eEV treatment with different inhibitors in H/R conditions. MEK1/2/eNOS/GC inhibitors abolished eEV-protective effect. *p* < 0.0001 NONE vs. eEV; *p* < 0.0001 MEK1/2/eNOS/GC inhibitors vs. eEV. Data were normalized to the mean value of normoxic control (CTRL N). (**d**) Infarct size in isolated rat hearts exposed to eEV and pretreated with different inhibitors. The protective effect of eEV (1 × 10^9^) was abrogated by pretreatment with the antagonist of MEK1/2, U0126 (*n* = 3), the inhibitor of eNOS, L-NNA (*n* = 3), and the blocker of GC, ODQ (*n* = 3). *p* < 0.01 eEV vs. eNOS inhibitor; *p* < 0.0001 eEV vs. MEK1/2 and GC inhibitors.

**Figure 6 cells-10-00013-f006:**
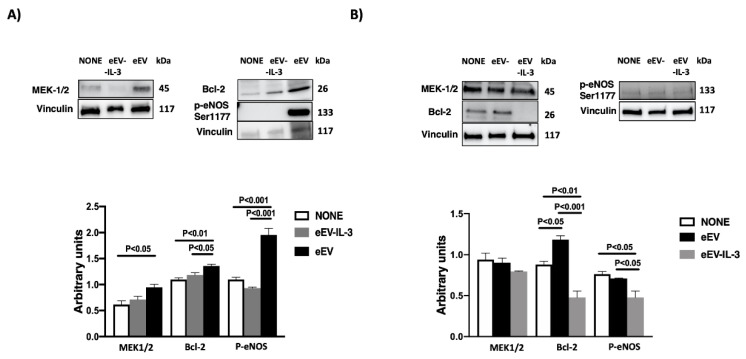
Effects of eEV and eEV-IL-3 on cardioprotective pathway. (**A**) Westen blot analysis of eEV and eEV-IL-3 in vitro treatment (transwell experiments). The expression of MEK1/2, p-eNOS, and B-cell-lymphoma-2 (Bcl-2) was evaluated in H9c2 cells exposed to H/R conditions and eEV or eEV-IL-3 treatment, normalized to vinculin. Untreated cells served as internal control (NONE) (*n* = 3). MEK1/2: *p* < 0.05 NONE vs. eEV (*n* = 4); Bcl-2: *p* < 0.05 eEV vs. eEV-IL-3; *p* < 0.01 NONE vs. eEV (*n* = 3); p-eNOS: *p* < 0.01 NONE vs. eEV and eEV vs. eEV-IL-3 (*n* = 3). (**B**) Western blot analysis of eEV and eEV-IL-3 in vitro treatment. The expression of MEK1/2, p-eNOS, and Bcl-2 was evaluated in H9c2 cells directly exposed to H/R conditions and eEV or eEV-IL-3 treatment, normalized to vinculin. Untreated cells served as internal control (NONE). Bcl-2: *p* < 0.001 eEV vs. eEV-IL-3; *p* < 0.05 NONE vs. eEV; *p* < 0.01 NONE vs. eEV-IL-3; p-eNOS: *p* < 0.05 eEV vs. eEV-IL-3; *p* < 0.05 NONE vs. eEV-IL-3 (*n* = 3).

**Figure 7 cells-10-00013-f007:**
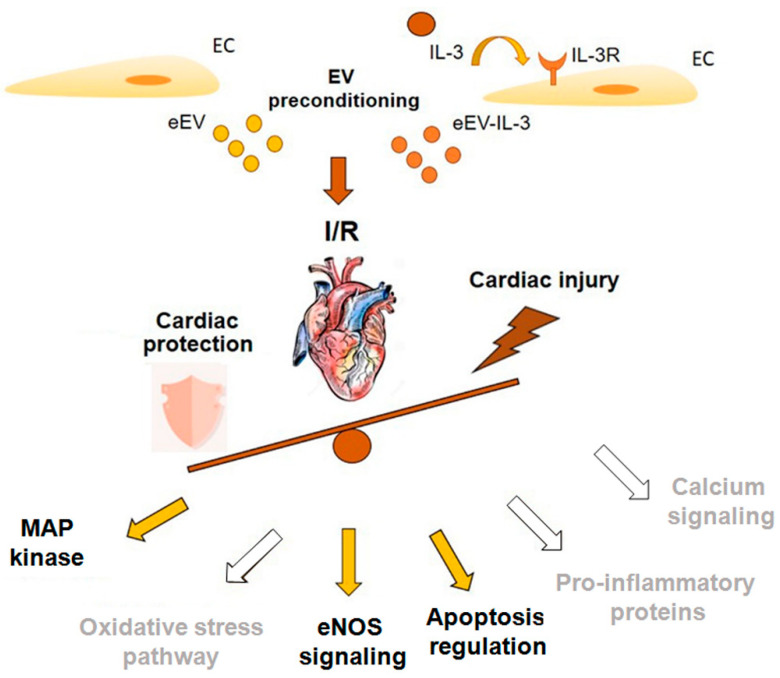
Schematic representation of eEV and eEV-IL-3 action on I/R injury. eEV and eEV-IL-3 were administered ex-vivo prior to I/R injury. A selective activation of specific signaling components likely mediates eEV cardioprotection. Different molecules can drive preconditioning protective signaling upon eEV treatment. eNOS, ERK1/2, and the anti-apoptotic protein Bcl-2 were validated. Other predicted but not validated pathways include PI3-kinase/Akt and oxidative stress (indicated by the gray font–white arrows). In response to eEV-IL-3, the transfer of inflammatory, apoptosis mediators and caveolin 1 likely change the balance and may trigger death signaling ex vivo.

**Table 1 cells-10-00013-t001:** List of enriched pathways in which proteins carried by eEV and eEV-IL-3 are involved.

Ingenuity Canonical Pathways	−log(*p*-Value)
Mitochondrial Dysfunction	12.9
Caveolar-Mediated Endocytosis Signaling	11.3
Integrin Signaling	9.69
Remodeling of Epithelial Adherens Junctions	9.26
Oxidative Phosphorylation	8.34
Regulation of eIF4 and p70S6K Signaling	8.02
Actin Cytoskeleton Signaling	7.97
Sirtuin Signaling Pathway	7.33
**NRF2-mediated Oxidative Stress Response**	6.14
**PI3K/AKT** Signaling	5.95
Regulation of Actin-based Motility by Rho	5.88
**VEGF** Signaling	5.05
p70S6K Signaling	4.82
**Leukocyte Extravasation Signaling**	4.41
**Apoptosis** Signaling	3.32
mTOR Signaling	3.16
**Hypoxia Signaling in the Cardiovascular System**	3.14
Clathrin-mediated Endocytosis Signaling	3.11
**Endoplasmic Reticulum Stress Pathway**	2.96
Protein Kinase A Signaling	2.75
Antigen Presentation Pathway	2.42
**ERK/MAPK** Signaling	2.34
Granulocyte Adhesion and Diapedesis	2.23
Cardiac Hypertrophy Signaling	2.05
Role of NFAT in Regulation of the Immune Response	1.98
CXCR4 Signaling	1.98
Glutathione Redox Reactions I	1.93
Acute Phase Response Signaling	1.92
Calcium Signaling	1.89
Glutathione-mediated Detoxification	1.74
**Arginine Biosynthesis IV**	1.64
Thioredoxin Pathway	1.51
Aspartate Degradation II	1.51
Calcium-induced T Lymphocyte Apoptosis	1.46
IL-1 Signaling	1.46
FcÎ^3^ Receptor-mediated Phagocytosis in Macrophages and Monocytes	1.44
**Nitric Oxide Signaling in the Cardiovascular System**	1.44
Superoxide Radicals Degradation	1.39
CCR3 Signaling in Eosinophils	1.31

**Table 2 cells-10-00013-t002:** Diseases and function annotation, performed by IPA analysis, obtained comparing eEV-IL-3 vs. eEV protein cargo.

Diseases or Functions Annotation	*p*-Value	Predicted Activation State	Activation z-Score	# Molecules
Cell movement of endothelial cells	4.09 × 10^9^	Increased	2.919	41
Migration of endothelial cells	9.55 × 10^9^	Increased	2.429	38
Vasculogenesis	2.45 × 10^8^	Increased	2.106	46
Interaction of endothelial cells	7.55 × 10^8^	Increased	2.587	22
Cell death of endothelial cells	1.21 × 10^7^		−1.535	21
Binding of endothelial cells	2.61 × 10^7^	Increased	2.402	21
Apoptosis of endothelial cells	6.80 × 10^7^		−1.591	19
Binding of vascular endothelial cells	5.83 × 10^6^	Increased	2.825	15
Adhesion of endothelial cells	7.63 × 10^6^	Increased	2.188	15
Apoptosis of vascular endothelial cells	8.03 × 10^6^		−0.969	14
Endothelial cell development	1.60 × 10^5^	Increased	2.57	30
Movement of vascular endothelial cells	2.68 × 10^5^		1.976	21
Adhesion of vascular endothelial cells	6.44 × 10^5^	Increased	3.087	11
Attachment of vascular endothelial cells	9.14 × 10^5^			4
Proliferation of endothelial cells	1.16 × 10^4^	Increased	2.363	26
Apoptosis of microvascular endothelial cells	3.45 × 10^4^		0.261	6
Migration of vascular endothelial cells	4.53 × 10^4^		1.604	17
Synthesis of reactive oxygen species	0.0008		−0.132	13
Cell spreading of endothelial cells	0.0014	Increased	2.219	5
Endothelial barrier function of vascular endothelial cells	0.0022			3
Cell viability of endothelial cells	0.0026		1.633	7
Formation of endothelial tube	0.0035			4
Generation of reactive oxygen species	0.0045		0.714	5
Production of reactive oxygen species	0.0050		−0.566	10
Morphology of endothelial cells	0.0068			3
Cell movement of muscle cells	0.0068			4
Cell spreading of vascular endothelial cells	0.0090			3
Survival of vascular endothelial cells	0.0095		1	5
Transendothelial migration of regulatory T lymphocytes	0.0097			2
Cell movement of muscle precursor cells	0.0097			2
Occlusion of artery	0.0138			5
Tubulation of endothelial cells	0.0147		0.632	10
Angiogenesis of endothelial cells	0.0147			3
Differentiation of vascular endothelial cells	0.0147			3
Transendothelial migration of T lymphocytes	0.0147			3
Coronary artery disease	0.0157			2
Adhesion of muscle cells	0.0157			2
Sliding of myofilaments	0.0173			5
Biosynthesis of hydrogen peroxide	0.0182			3
Atherosclerosis	0.0269			4
Differentiation of endothelial cells	0.0285		0.447	5
Migration of endothelial progenitor cells	0.0312			2
Proliferation of myoblasts	0.0312			2
Cell proliferation of vascular endothelial cells	0.0323	Increased	2.735	12
Shape change of vascular endothelial cells	0.0371		0.555	5
Permeability of endothelial progenitor cells	0.0413			1
Injury of cardiomyocytes	0.0413			1
Diastolic heart failure	0.0413			1
Perfusion of myocardium	0.0413			1
Vasoconstriction of artery	0.0413			1
Arrest in mid-G1 phase of microvascular endothelial cells	0.0413			1
Anoikis of vascular endothelial cells	0.0413			1
Delay in initiation of fusion of myoblasts	0.0413			1
Aggregation of myoblasts	0.0413			1
Activation of myoblasts	0.0413			1
Activation of myotube	0.0413			1
Morphology of cardiovascular system	0.0423			6

**Table 3 cells-10-00013-t003:** Inflammatory pathways and list of proteins modulated in eEV-IL-3 with respect to eEV. Bold pathways are statistically significant.

Ingenuity Canonical Pathways	−log(*p*-Value)	Gene List
**Agranulocyte Adhesion and Diapedesis**	4.76	ACTA1, ACTG1, CD99, CDH5, FN1, GLG1, GNAI2, ICAM1, ICAM2, ITGA2, ITGA5, ITGA6, ITGB1, MSN, MYH10, MYH9, MYL6, PECAM1, PODXL, RDX
**Leukocyte Extravasation Signaling**	4.41	ACTA1, ACTG1, ACTN1, ACTN4, CD99, CDH5, CTNNA1, CTNNB1, F11R, GNAI2, ICAM1, ITGA2, ITGA5, ITGA6, ITGB1, ITGB3, MAPK1, MSN, MYL6, PECAM1, PXN, RDX
**NF-kB Activation by Viruses**	2.24	ITGA2, ITGA5, ITGA6, ITGAV, ITGB1, ITGB3, MAPK1, RALA, RALB, RAP2B
**Granulocyte Adhesion, and Diapedesis**	2.23	CD99, CDH5, GLG1, GNAI2, ICAM1, ICAM2, ITGA2, ITGA5, ITGA6, ITGB1, ITGB3, MSN, PECAM1, RDX
**IL-8 Signaling**	2.01	ANGPT2, CSTB, GNAI2, GNB1, GNG12, ICAM1, IQGAP1, ITGAV, ITGB3, LASP1, MAP2K1, MAPK1, RALA, RALB, RAP2B, RHOC
**Role of NFAT in Regulation of the Immune Response**	1.98	CALM1, CHP1, GNA11, GNAI2, GNAQ, GNB1, GNG12, HLA-A, HLA-B, MAP2K1, MAPK1, RALA, RALB, RAP2B, XPO1
**Acute Phase Response Signaling**	1.92	A2M, CP, FGA, FN1, HMOX2, HNRNPK, HP, MAP2K1, MAPK1, RALA, RALB, RAP2B, RBP4, SERPINE1
**Complement System**	1.81	C1QBP, C6, C8B, CD59, MASP1
**IL-1 Signaling**	1.46	GNA11, GNAI2, GNAQ, GNB1, GNG12, MAPK1, PRKAR1A, PRKAR2A
CCR3 Signaling in Eosinophils	1.31	CALM1, CFL1, GNAI2, GNB1, GNG12, MAP2K1, MAPK1, RALA, RALB, RAP2B
Regulation of IL-2 Expression in Activated and Anergic T Lymphocytes	1.21	CALM1, CHP1, MAP2K1, MAPK1, RALA, RALB, RAP2B
Role of NFAT in Cardiac Hypertrophy	1.18	CALM1, CHP1, GNAI2, GNAQ, GNB1, GNG12, MAP2K1, MAPK1, PDIA3, PRKAR1A, PRKAR2A, RALA, RALB, RAP2B
IL-12 Signaling and Production in Macrophages	0.93	APOB, APOC2, CLU, MAP2K1, MAPK1, MST1, PCYOX1, PON1, RBP4
IL-3 Signaling	0.78	CHP1, MAP2K1, MAPK1, RALA, RALB, RAP2B
OX40 Signaling Pathway	0.76	B2M, HLA-A, HLA-B, TNFSF4
IL-2 Signaling	0.75	MAP2K1, MAPK1, RALA, RALB, RAP2B
IL-4 Signaling	0.73	HLA-A, HLA-B, HMGA1, RALA, RALB, RAP2B
GM-CSF Signaling	0.61	MAP2K1, MAPK1, RALA, RALB, RAP2B
IL-15 Signaling	0.58	MAP2K1, MAPK1, RALA, RALB, RAP2B
IL-17 Signaling	0.47	MAP2K1, MAPK1, RALA, RALB, RAP2B
IL-6 Signaling	0.31	A2M, MAP2K1, MAPK1, RALA, RALB, RAP2B
NF-kB Signaling	0	RALA, RALB, RAP2B, UBE2N, UBE2V1
CD40 Signaling	0	ICAM1, MAP2K1, MAPK1

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
