# Peer review of "The Inflammatory Cytokine IL-3 Hampers Cardioprotection Mediated by Endothelial Cell-Derived Extracellular Vesicles Possibly via Their Protein Cargo"

_cells, 2020, doi:10.3390/cells10010013_

Round 1

Reviewer 1 Report

Penna et al. in this paper analyzed the role of IL-3 preconditioning in the cargo and release of Endothelial Extracellular vesicles. 

Despite the idea is interesting there are some points that have to be clarified.

Major points:

-In the extracellular vesicles characterization the authors should provide information regarding the quality of their EVs preparation. A negative markers for EVs should be add in the wester blot analysis.

-The paper investigate the role of endothelial EVs in cardioprotection. I don't understand why all the in vitro experiments are performed in co-culture. The author should treat directly H9C2 with EVs in order to mimic the in vivo experiments. In this in vitro setting what is mimed is the role of the secreting factor (not only EVs but also MVs and soluble proteins) of the Microvascular endothelial cells after the treatment with or without of EVs/EVs-IL3 from Human umbilical vein endothelial cell.
All the in vitro experiments should be repeted with the appropriate setting, otherwise the conclusions regarding the expression of MEK1/2 and the phosphorylation of eNOS can't be associated directly to HUVEC EVs treatment.

-figure 4D statistic should be add to the figure in order to say that protein are enriched.

Minor point:
-In the discussion some piece of the text are in a different font.

Author Response

We thank the Reviewer for his/her appreciation of our experimental approach and positive comments.

In the extracellular vesicles characterization the authors should provide information regarding the quality of their EVs preparation. A negative markers for EVs should be add in the wester blot analysis.

As kindly suggested by the Reviewer the negative marker for EV (GM130) has been included in Fig 1B.

The paper investigate the role of endothelial EVs in cardioprotection. I don't understand why all the in vitro experiments are performed in co-culture. The author should treat directly H9C2 with EVs in order to mimic the in vivo experiments. In this in vitro setting what is mimed is the role of the secreting factor (not only EVs but also MVs and soluble proteins) of the Microvascular endothelial cells after the treatment with or without of EVs/EVs-IL3 from Human umbilical vein endothelial cell. All the in vitro experiments should be repeated with the appropriate setting, otherwise the conclusions regarding the expression of MEK1/2 and the phosphorylation of eNOS can't be associated directly to HUVEC EVs treatment.

As suggested by the Reviewer a dose response curve of EV treatment was performed in H9c2 cells and reported as Figure 2B,C. eEV at the highest dose (1*106 EV/cell), unlike eEV-IL-3, protected H9c2 cells against H/R damage. In Figure 6B the expression of MEK1/2, Bcl-2 and the phosphorylation of eNOS are reported. A comment has been also included in the Result section (line 423-435).

Figure 4D statistic should be add to the figure in order to say that protein are enriched

As kindly suggested by the Reviewer statistic has been included in Figure 4D.

In the discussion some piece of the text are in a different font.

According with the Reviewer comment, the font has been fixed.

Reviewer 2 Report

Manuscript cells-1030467 written by Penna et al. brings data about the potential confounding role of the pro-inflammatory cytokine IL-3 in the cardioprotection mediated by extracellular vesicles (EVs) released in vitro from endothelial cells (HUVECs). The effects of these EVs were examined in both in vitro and ex vivo experimental models of cardiac I/R injury. To my best knowledge, the paper is novel and brings original knowledge in the field. The topic of the paper is actual since the role of EVs in heart health, disease and cardioprotection is a hot topic of current cardiovascular science, thus paper might be of potential importance and interest of readers. Paper is well written and is of a good quality, thus is suitable for publication in Cells. However, I have several comments to the paper:

  • In Introduction authors should explain why they decided to reveal effects of particularly IL-3 and no other cytokines. Were there some preliminary experiments with more candidates? Is there any literature-based ratio for IL-3? I expect that IL-3 was not selected randomly so the choice should be clearly explained.
  • In Introduction when authors introduce potential role of inflammation in myocardial I/R injury I suggest cite also the recent comprehensive review article assessing role of cytokines and inflammation in heart disease (PMID 29862462) in addition to the older review paper by Frangogiannis (Reference no.9) cited in the paper.
  • In Methods, to make the paper convenient to readers, I suggest briefly explain the protocol of TEM, similarly as in the case of other methodologies included in the paper, not only cite the previous papers using this method.
  • In Methods, Western blotting, please add information about secondary antibodies and method of visualization (ECL?)
  • In Results, surprisingly, proteins of IL-3 pathway were not modulated in IL-3-eEVs in comparison to intact eEVs (Table 3). How authors explain this finding? I would expect that IL-3 pathway will be affected by exposition to IL-3.
  • In Statistics, it should be clearly defined when ANOVA and when T-test were used since these are suitable for different types of comparisons (T-test for two groups only and ANOVA for more than two groups and should be followed by adequate post-hoc test).
  • In whole paper, many abbreviations, e.g. NTA, FACS, L-NNA, ODQ, CAN, TFA, GO etc., are not properly introduced when firstly appear in the text. In contrary, some abbreviations are introduced twice in the same paragraph (e.g. FBS in paragraph 2.1). Please, carefully check all abbreviations and their explanations. In line, I suggest create the List of Abbreviations for the paper.

Author Response

We thank the Reviewer for his/her appreciation of our experimental approach and positive comments.

In Introduction authors should explain why they decided to reveal effects of particularly IL-3 and no other cytokines. Were there some preliminary experiments with more candidates? Is there any literature-based ratio for IL-3? I expect that IL-3 was not selected randomly so the choice should be clearly explained.

We thank the Reviewer for this comment. As suggested an explanation for the selection of IL-3 has been included in the Introduction section (line 86-94)

In Introduction when authors introduce potential role of inflammation in myocardial I/R injury I suggest cite also the recent comprehensive review article assessing role of cytokines and inflammation in heart disease (PMID 29862462) in addition to the older review paper by Frangogiannis (Reference no.9) cited in the paper.

As kindly suggested by the Reviewer the indicated paper has been included in the present version of the Ms.

In Methods, to make the paper convenient to readers, I suggest briefly explain the protocol of TEM, similarly as in the case of other methodologies included in the paper, not only cite the previous papers using this method.

As kindly suggested by the Reviewer detailed information onTEM protocol has been included in the present version of the Ms (line 143-149).

In Methods, Western blotting, please add information about secondary antibodies and method of visualization (ECL?)

As kindly suggested by the Reviewer, the present version of the Ms includes detailed information of western blot analysis (line 256-257).

In Results, surprisingly, proteins of IL-3 pathway were not modulated in IL-3-eEVs in comparison to intact eEVs (Table 3). How authors explain this finding? I would expect that IL-3 pathway will be affected by exposition to IL-3.

We thank the Reviewer for this comment. As expected by the Reviewer mass spectrometry analysis identify proteins involved in IL-3 signalling, however, bioinformatic analysis failed to detect statistical significance in the enriched pathways. In particular, protein cluster in the IL-3 signalling cascade includes: CHP1, MAP2K1, MAPK1, RALA, RALB, RAP2B (Table 3). From among these proteins, only MAP2K1 was not enriched in eEV-IL-3. In addition, proteins involved in the regulation of apoptosis and cell cycle have been detected among enriched proteins in eEV-IL-3 by mass spectrometry analysis (Supplementary Table S1). Actually, both pathways are triggered by IL-3.

In Statistics, it should be clearly defined when ANOVA and when T-test were used since these are suitable for different types of comparisons (T-test for two groups only and ANOVA for more than two groups and should be followed by adequate post-hoc test).

As kindly requested by the Reviewer in the present version of the Ms Statistics have been clearly described (line 264-268).

In whole paper, many abbreviations, e.g. NTA, FACS, L-NNA, ODQ, CAN, TFA, GO etc., are not properly introduced when firstly appear in the text. In contrary, some abbreviations are introduced twice in the same paragraph (e.g. FBS in paragraph 2.1). Please, carefully check all abbreviations and their explanations. In line, I suggest create the List of Abbreviations for the paper.

All abbreviations have been included in the present version of the Ms (line 41-73).

Round 2

Reviewer 1 Report

No further comments.